# Effect of Selol on Tumor Morphology and Biochemical Parameters Associated with Oxidative Stress in a Prostate Tumor-Bearing Mice Model

**DOI:** 10.3390/nu16172860

**Published:** 2024-08-27

**Authors:** Małgorzata Sochacka, Grażyna Hoser, Małgorzata Remiszewska, Piotr Suchocki, Krzysztof Sikora, Joanna Giebułtowicz

**Affiliations:** 1Department of Drug Chemistry, Pharmaceutical and Biomedical Analysis, Faculty of Pharmacy, Medical University of Warsaw, 1 Banacha Street, PL-02097 Warsaw, Poland; piotr.suchocki@wum.edu.pl (P.S.); joanna.giebultowicz@wum.edu.pl (J.G.); 2Department of Translational Immunology and Experimental Intensive Care, Centre of Postgraduate Medical Education, Ceglowska 80, PL-01809 Warsaw, Poland; 3Department of Pharmacology, National Medicines Institute, 30/34 Chełmska Street, PL-00725 Warsaw, Poland; malgorzata.remiszewska@nil.gov.pl; 4Pathomorphology Centre, National Medical Institute of the Ministry of the Interior and Administration, 137 Wołoska Street, PL-02507 Warsaw, Poland

**Keywords:** cancer, selenium, Selol, antioxidants, tumor

## Abstract

Prostate cancer is the leading cause of cancer death in men. Some studies suggest that selenium Se (+4) may help prevent prostate cancer. Certain forms of Se (+4), such as Selol, have shown anticancer activity with demonstrated pro-oxidative effects, which can lead to cellular damage and cell death, making them potential candidates for cancer therapy. Our recent study in healthy mice found that Selol changes the oxidative–antioxidative status in blood and tissue. However, there are no data on the effect of Selol in mice with tumors, considering that the tumor itself influences this balance. This research investigated the impact of Selol on tumor morphology and oxidative–antioxidative status in blood and tumors, which may be crucial for the formulation’s effectiveness. Our study was conducted on healthy and tumor-bearing animal models, which were either administered Selol or not. We determined antioxidant enzyme activities (Se-GPx, GPx, GST, and TrxR) spectrophotometrically in blood and the tumor. Furthermore, we measured plasma prostate-specific antigen (PSA) levels, plasma and tumor malondialdehyde (MDA) concentration as a biomarker of oxidative stress, selenium (Se) concentrations and the tumor ORAC value. Additionally, we assessed the impact of Selol on tumor morphology and the expression of p53, BCL2, and Ki-67. The results indicate that treatment with Selol influences the morphology of tumor cells, indicating a potential role in inducing cell death through necrosis. Long-term supplementation with Selol increased antioxidant enzyme activity in healthy animals and triggered oxidative stress in cancer cells, activating their antioxidant defense mechanisms. This research pathway shows promise in understanding the anticancer effects of Selol. Selol appears to increase the breakdown of cancer cells more effectively in small tumors than in larger ones. In advanced tumors, it may accelerate tumor growth if used as monotherapy. Therefore, further studies are necessary to evaluate its efficacy either in combination therapy or for the prevention of recurrence.

## 1. Introduction

Prostate cancer (PCa) is the most prevalent cancer in elderly men, characterized by its aggressive nature leading to metastasis and, frequently, fatality [1]. It is currently the second leading cause of death for men in Western societies [2]. Prostate cancer’s pathogenesis remains unclear. PCa development and progression involve various factors, including aging, environmental factors, lifestyle choices, physical activity, genetic alterations, and hormonal influences. Oxidative stress (OS) is thought to contribute directly to the development of prostate cancer [3]. Therefore, OS represents a potential therapeutic target for addressing prostate conditions such as prostatic hypertrophy, cancer, or chronic prostatitis. Oxidative stress in cells results from an imbalance between oxidants and antioxidants, leading to damaged lipids, proteins, and DNA structures. Experimental data suggest that cancer cells, as a result of genetic mutations, have reduced activity in certain antioxidant enzymes compared with normal cells [4]. This diminished enzymatic activity can induce oxidative stress, leading to damage to cellular components and structures in cancerous cells. Among the diverse strategies employed in cancer treatment, one approach involves deliberately elevating reactive oxygen species (ROS) levels within cancer cells. Chemotherapeutic agents like cisplatin, arsenic trioxide, and anthracycline antibiotics operate through this mechanism [5]. 

Selenium (Se), a micronutrient, participates in various physiological processes [6]. Its importance lies in its role in the catalytic centers of antioxidant enzymes, including selenium-dependent glutathione peroxidase (Se-GPx) and thioredoxin reductase (TrxR) [7]. Selenite, in erythrocytes, when interacting with glutathione, forms biologically active selenodiglutathione, with potent anticancer properties that induce cancer cell apoptosis. Inorganic selenium compounds (+4), more than organic ones (+2), have been found to inhibit cancer cell growth and proliferation due to their pro-oxidant properties, particularly at doses exceeding therapeutic levels [8]. This phenomenon presents potential for anticancer therapy; however, the high toxicity of commercially available inorganic Se (+4) compounds limits their applicability. As a result, there is an ongoing search for selenium (+4) compounds that combine high chemoactivity with low toxicity. Consequently, the synthesis of novel chemopreventive compounds is crucial, with a focus on evaluating their efficacy against various cancer types. 

Selol is a selenintriglyceride compound derived from sunflower oil. It has lower toxicity compared with sodium selenite (+4) and has no mutagenic effects [9]. In addition, Selol shows strong cytostatic activity against cancer cell lines, with no side effects on normal cells. Selol is a formulation with potential anticancer activity. Previous in vitro studies demonstrated Selol’s significant antitumor effects on human HL-60 leukemia cells, including drug-resistant variants (HL-60/Dox and HL-60/Vinc) [10], as well as HeLa [11] and Caco-2 cancer cells [12]. Książek et al. observed that prostate cancer cells, LNCaP, were more susceptible to apoptosis induction caused by the presence of ROS compared with normal prostate cells—PNT1A [13]. The safety of Selol for normal cells was confirmed in PC12 cells. Selol, through the regulation of free radical levels, the enhancement of the antioxidant system, and the inhibition of apoptosis, protects against oxidative damage and death induced by SNP. Selol is believed to induce the production of ROS in cancer cells, leading to excessive oxidative stress. Healthy cells can manage this stress, while in cancer cells, it may result in apoptosis [14]. 

Under conditions of chronic oxidative stress, cells activate defensive mechanisms, primarily through the activation of phase 2 enzymes via the Nrf2/ARE signaling pathway. In vivo experiments conducted on a healthy animal model have shown that Selol significantly influences changes in non-enzymatic antioxidants (thiols) and the intracellular and extracellular redox status [15]. 

However, these findings have not been validated in a cancer animal model, where the tumor itself affects the organism’s redox status. Considering the physical conditions, our previous study showed that treating mice with Selol resulted in a drop in the rate of body mass reduction and stopped the increase in plasma prostatic specific antigen (PSA) levels. The expression of genes, involved in oxidative stress following treatment with Selol, in LNCaP cells, did not change. However, it is important to remember that gene expression is not directly linked to enzyme activity. Many factors influence the formation of an active protein [16]. 

Our goal was to investigate the impact of long-term Selol administration on tumor growth and the oxidative–antioxidative status in both blood and the tumor, which could be crucial for the formulation’s effectiveness. We assessed the levels of the antioxidant enzymes glutathione S-transferase (GST), glutathione peroxidase (GPx), and thioredoxin reductase (TrxR), regulated by the Nrf2/ARE pathway, in both the blood and tumor of mice with xenografted LNCaP prostate cancer. Additionally, the study considered the marker of oxidative damage malondialdehyde (MDA) and tumor oxygen-radical absorbance capacity (ORAC) and determined the selenium levels. Prostate-specific antigen (PSA) concentrations before and after Selol administration were also determined. Therefore, it was decided to perform a macroscopic examination of the results obtained and analyze the alterations in the tumor at the histopathological level. Histopathological examinations like morphology and p53, BCL2, and Ki-67 expression were conducted. 

## 2. Materials and Methods

### 2.1. Compound Characterization

The synthesis of Selol was carried out in the Department of Drugs Analysis at Warsaw Medical University (Patent Pol. PL 176,530 (Cl. A61K31/095)). Selol is a mixture of selenitriglicerides obtained by the chemical modification of sunflower oil, in which a minimum of 11 distinct selenium-containing triglycerol derivatives were identified by using mass spectrometry [17]. In the experiments, Selol 5% at a dose of 17 mg Se/kg body mass was used. 

### 2.2. Ethics Statement

All animal experiments were conducted in accordance with the guidelines set forth by European Communities Directive 2010/63/EU. Ethical approval for the study was obtained from the IV Local Ethics Committee for Animal Experimentation in Warsaw, Poland, under protocol number 33/2009 dated 1 April 2009. Every effort was undertaken to minimize animal distress and to limit the number of animals used. The animal experiments adhered to the ARRIVE guidelines. 

### 2.3. Animal Model

Adult, immunodeficient, sexually mature male NSG mice (NOD.Cg-Prkdc/sc dIl2rg), (approximately 27–35 g, 12 weeks old) were purchased from Charles River Laboratories (Germany). Animals were housed in cages (Centre for Postgraduate Medical Training in Warsaw) maintained under controlled environmental conditions at 20 ± 2 °C room temperature, 40 ± 5% relative humidity, and a 12 h light/dark cycle with a dawn/dusk effect. Mice were fed a complete feed mixture for laboratory animals (LSM, Agropol, Lublin, Poland) and had access to water ad libitum. To reduce stress, the animals were handled for 10 minutes each day and acclimated to the oral gavage procedure over a period of one week prior to the start of the experiments.

### 2.4. Experimental Protocol

The induction of the tumor was performed in accordance with the results of the pilot studies. The procedure involved implanting 5 million LNCaP cells (an epithelial cell line derived from human prostate carcinoma) into the shoulder of mice. Cells were suspended in Matrigel (Corning^®^ Matrigel^®^ Matrix GFR PhenolRF Mouse; Sigma-Aldrich, Sant Louis, MI, USA), a vital protein mixture that promotes optimal cell culture growth, minimizes single-cell dispersion, and contains essential growth factors.

The animal study lasted eight weeks. Tumors were induced in healthy NSG mice (*n* = 12). At the end of the fifth week after LNCaP cell inoculation, a significant darkening of the skin at the injection site and the development of tumors were observed. Then, the mice were randomly assigned to the following four main groups: (1) Ca—mice with xenografted LNCaP prostate cancer (*n* = 6); (2) Control—mice without prostatic tumors (*n* = 5); (3) Control + Se—mice without prostatic tumors treated with Selol (*n* = 5); (4) Ca + Se—mice with xenografted LNCaP prostate cancer treated with Selol (*n* = 6). Each of the study groups (Control + Se and Ca + Se) was supplemented daily per os with a single dose of Selol diluted with vegetable oil equivalent to 17 mg Se/kg body weight (which is approx. 20% of LD50—dose based on the in vivo results from the Selol toxicity study; unpublished data). The Control and Ca groups were fed the standard diet with the same rate of pure vegetable oil (placebo) as the study groups. Placebo and Selol were administered to the animals over three weeks. During the treatment period, all mice were weighed and observed for changes in their behavior twice a week. At the end of the experiment, mice were anesthetized with halothane, the blood was collected for the biochemical measurements, and the tumors were isolated from each mouse for further investigation. In the fifth and eighth weeks of the experiment, PSA concentrations were determined to confirm the presence of prostate cancer and/or the effect of Selol supplementation on the marker.

### 2.5. Biochemical Analysis 

The animals underwent a 12-h fast before being sacrificed. Blood samples were collected into heparinized tubes and centrifuged at 1000× *g* for 15 min at 4 °C to obtain plasma. Red blood cells were washed twice with 0.9% NaCl, and the plasma was refrigerated and stored at −80 °C until analysis. On the analysis day, the samples were thawed at room temperature. Red blood cells intended for enzymatic analysis were hemolyzed by using an equal volume of 3 mM phosphate buffer at pH 7.4 containing 1 mM EDTA. The hemolysates were centrifuged at 1000× *g* for 20 min at 4 °C, and the supernatants were used for further measurements. The tumors were extracted, weighed, and divided into small fragments. The tumor fragments were briefly exposed to liquid nitrogen for preservation before being stored at −80 °C. All tumor samples were processed within a two months. Before measurement, the tumors were homogenized by using a manual glass homogenizer. For enzyme activity measurements, the samples were homogenized in a chilled medium consisting of 5 mM phosphate buffer, 0.25 mM sucrose, and 0.5 mM EDTA at pH 7.2. Meanwhile, for MDA measurements, a cold medium containing 5 mM pyrophosphate buffer at pH 7.4 was used. The cytosolic fraction of the homogenates was separated by centrifugation at 10,000× *g* for 20 min at 4 °C.

The following parameters were determined in plasma, erythrocytes, and tumor homogenate supernatants: selenium-dependent glutathione peroxidase (Se-GPx) and total glutathione peroxidase (GPx) activities, and thioredoxin reductase (TrxR) and glutathione S-transferase (GST) activities. Furthermore, plasma prostate-specific antigen (PSA) levels, tumor ORAC values (antioxidant capacity), concentrations of malondialdehyde (MDA) in both plasma and tumors (a marker of lipid peroxidation), and selenium concentrations in erythrocytes and tumors were assessed.

#### 2.5.1. Determination of Enzyme Activity

Selenium-dependent glutathione peroxidase (Se-GPx) and glutathione peroxidase (GPx) activities in plasma, red blood cells, and tumor homogenates were determined spectrophotometrically at a wavelength of 340 nm, using a method originally developed by Paglia and Valentine, modified by Wendel [18,19]. The reaction was carried out at 25 °C in 50 mM sodium phosphate buffer containing 0.40 mM EDTA at pH 7.0. A supernatant/plasma volume of 10 µL was utilized for enzyme activity analysis. In the final reaction mixture (total volume of 220 µL), the concentrations were as follows: reduced glutathione (GSH) at 1.0 mM, NADPH at 65 µM, and sodium azide at 0.17 mM. For the Se-GPx activity assay, a substrate tert-butyl hydroperoxide was employed at a concentration of 0.02 mM. Cumene hydroperoxide at 1.05 mM concentration was used for the GPx activity determination.

Glutathione S-transferase (GST) activity in plasma, red blood cells, and tumor homogenates was quantified by using spectrophotometric methods at a wavelength of 340 nm, employing the Habig assay [20]. The enzymatic reaction took place at 25 °C in 50 mM sodium phosphate buffer supplemented with 0.50 mM EDTA at pH 7.5. A 10 µL aliquot of supernatant/plasma was utilized to assess enzyme activity. The final reaction mixture, with a total volume of 200 µL, contained 2 mM reduced glutathione (GSH) and 1 mM 1-chloro-2,4-dinitrobenzene (CDNB) as the substrate. 

Thioredoxin reductase (TrxR) activity in plasma, red blood cells, and tumor homogenates was assessed spectrophotometrically at a wavelength of 412 nm, using the modifications outlined by Hill et al. [21]. The enzymatic reaction was conducted at 37 °C in 50 mM sodium phosphate buffer supplemented with 1 mM EDTA at pH 7.0. A 10 μL of the supernatant was used for the enzyme activity analysis. In the final reaction mixture, with a volume of 200 μL, the concentrations were as follows: 4 mM 5,5′-dithiobis(2-nitrobenzoic acid) (DTNB) as the substrate, 2 μM nicotinamide adenine dinucleotide phosphate (NADPH) as the enzymatic reaction cofactor, and 1 mM auranofin (ATM) as a specific inhibitor of the enzyme under study. Thioredoxin reductase activity was determined by measuring the difference in enzymatic activity between samples tested without and with the inhibitor [22,23]. 

#### 2.5.2. Determination of Malondialdehyde Concentration

The concentration of malondialdehyde (MDA) in both plasma and tumor homogenates was quantified by using an ELISA spectrophotometric assay kit (Wuhan EIAAB Science Co., Ltd., Wuhan, China), following the manufacturer’s instructions.

#### 2.5.3. Determination of Oxygen-Radical Absorbance Capacity (ORAC)

The ORAC-FL assay, as described by Ou et al. [24], involved measuring the antioxidant capacity of samples. Hitachi F-7000 (Hitachi, Tokyo, Japan) spectrofluorometer with excitation at 485 nm and emission at 520 nm was used to determine the ORAC value of supernatants of tumor tissues. Black 96-well plates (Greiner Bio-One, Kremsmünster, Austria) were employed in the assay. All solutions (fluorescein and AAPH) were freshly prepared in PBS buffer at pH 7.4 daily. The reaction mixture consisted of 13 mM AAPH and fluorescein at 40 nM.

#### 2.5.4. Determination of Prostate-Specific Antigen (PSA) Concentration

The PSA levels in the mice’s plasma were assessed twice, labeled as PSA1 and PSA2. At the end of the fifth week, 150 µL of blood was extracted from the tail of each mouse for the initial PSA measurement (PSA1). After three weeks of administering Selol/placebo, the mice were humanely sacrificed, and their blood was gathered for the subsequent PSA level analysis (PSA2). Following collection, the blood was centrifuged to obtain plasma for the evaluation of PSA levels by using immunodetection (Human PSA-total ELISA Kit; Sigma-Aldrich, Sant Louis, MI, USA; as recommended by the manufacturer). 

#### 2.5.5. Determination of Protein and Hemoglobin Concentration

The protein concentration in tumor tissue supernatants was determined by using a spectrophotometric assay with Bradford reagent (Sigma-Aldrich, Sant Louis, MI, USA). The absorbance of the protein-bound Coomassie Brilliant Blue G-250 dye was read at 595 nm. Protein concentrations were calculated based on a standard curve prepared by using bovine serum albumin (BSA) as the reference standard. The hemoglobin concentration of red blood cell (RBC) hemolysates was assessed spectrophotometrically at 546 nm by using a standard immunodetection assay (Human hemoglobin ELISA Kit**;** Sigma-Aldrich, Sant Louis, MI, USA; as recommended by the manufacturer). Measurements of enzymatic activity, hemoglobin concentration, and protein absorbance were conducted by using a spectrophotometer microplate reader (Synergy MX, BioTek^®^ Instruments, Inc., Winooski, VT, USA).

#### 2.5.6. Determination of Selenium Concentration

To quantify the complete selenium (Se) content in red blood cells and tumor tissues, an inductively coupled plasma mass spectrometer (ICP-MS) (Thermo Fisher Scientific, Waltham, MA, USA) was used [25,26]. Tissue samples were homogenized in a mixture of 65% HNO_3_ and 30% H_2_O_2_ (in a 3:1 ratio). Subsequently, homogenized samples were transferred to Teflon crucibles and subjected to mineralization by using a microwave mineralizer. For erythrocyte samples, 100 μL aliquots were transferred to Teflon crucibles containing a mineralization mixture as mentioned above, followed by mineralization. The samples were then used for ICPMS analyses, with quality control samples. 

### 2.6. Histopathological Examination

Histopathological evaluation was performed on tissue sections from tumor-bearing animals (*n* = 5). The lesions assessed were nodules with macroscopic diameters ranging from 3 to 4 mm in control animals and from 3 to 5 mm in animals treated with the active substance. After evaluation, tissues were processed by embedding in 10% buffered formalin, sectioned into 7 µm slices, and stained with hematoxylin and eosin (H&E). Examination of the prepared slides was conducted by using an, Olympus BX41 microscope with an Olympus DP25 camera and cellSens software For immunohistochemical evaluation, slides were cut to a thickness of 4 µm on salinized slides and transferred to a hothouse (50 °C). Staining was performed with DAKO antibodies by using Dako AutostainerLink 48 platform. Slides were transferred to buffer (Target Retrieval Solution EnVision FLEX) at pH 9.0 or 6.0 at PTLink to open antigenic dominance EnVision Detection Kit (Env FLEX, High pH, DAKO) was used. Antibodies against p53, BCL2, and Ki-67 (DAKO Omnis, Agilent Technologies, Santa Clara, CA, USA were used to assess protein expression levels and to evaluate tumor characteristics. Slides were stained with hematoxylin, washed with water, dehydrated with a series of alcohols, overexposed with xylene, and sealed in BDX.

### 2.7. Statistical Analysis

The data are presented as the means ± standard deviation (SD). Statistical analyses were performed by using one-way ANOVA to compare means among multiple groups, followed by Tukey’s and Dunnett’s post-hoc tests for pairwise comparisons. The Mann–Whitney U test was used for comparing two independent groups, and the Spearman correlation and multiple regression test was used to assess the strength and direction of the association between two and more than two variables, respectively. A *p*-value of less than 0.05 was considered significant. Statistical tests were performed by using Statistica software (version 10, StatSoft, TIBCO Software, Warsaw, Poland).

## 3. Results

### 3.1. Antioxidant Enzymes

The levels of antioxidant enzymes, such as Se-GPx, GST, and TrxR, in blood plasma were significantly higher in the group of mice with xenografted LNCaP prostate cancer (Ca) compared with the control group (Control) (*p* = 0.0001, *p* = 0.0061, and *p* = 0.02; Table 1). However, no significant differences were observed in the activities of antioxidant enzymes (Se-GPx, GST, TrxR, and GPx) in the erythrocytes of these two groups of mice. In the healthy mice group, the administration of Selol increased Se-GPx activity in blood plasma (Control + Se) (*p* = 0.0020), as well as Se-GPx, GPx, and GST activities in erythrocytes (*p* = 0.0003, *p* = 0.0004, and *p* = 0.0152, respectively; Table 1).

This effect was not observed in mice with xenografted LNCaP prostate cancer. However, in this group, the activities of Se-GPx, GPx, and GST were higher in the tumor compared with the mice receiving a placebo (Figure 1a,b).

### 3.2. Marker of Lipid Peroxidation

The plasma MDA levels were significantly higher in the group with LNCaP prostate cancer compared with the control group (*p* = 0.0037; Table 2). However, Selol administration to the healthy group increased MDA levels (*p* = 0.0065); this effect was not observed in mice with xenografted LNCaP prostate cancer. There were no significant differences observed in the concentration of MDA within the tumor, whether the mice were administered selenium or not (Figure 1c).

### 3.3. ORAC Value

The tumor tissue oxygen-radical absorbance capacity (ORAC) was significantly higher in the group of mice treated with Selol (Ca + Se) compared with the placebo group (Ca) (Figure 1d).

### 3.4. Selenium Concentration

The administration of Selol resulted in an increase in selenium levels in erythrocytes, both in the group of healthy mice and mice with LNCaP prostate cancer (*p* = 0.0001 in both cases). Interestingly, in mice receiving Selol, higher selenium concentrations were observed in LNCaP mice than in healthy mice (*p* = 0.0002; Table 2). As expected, the tumors in mice with xenografted LNCaP prostate cancer that received Selol (Ca + Se) showed significantly higher selenium levels compared with the control group that received the placebo (Ca) (Figure 1e).

### 3.5. Morphological Study

The tumors of both the study and placebo mice displayed a range of sizes, all characterized by noticeable blood vessel proliferation. Significant differences in tumor appearance were observed between the two experimental groups. Tumors in mice treated with Selol (Ca + Se) appeared darker in color, exhibited swollen interiors, and in some regions, had a “jelly-like” consistency. In contrast, tumors in mice treated with the placebo (Ca) maintained a dense and firm structure. Furthermore, there were no observable alterations in the tissue appearance of other organs in mice that received Selol (Ca + Se and Control + Se).

### 3.6. PSA Concentration, Tumor Mass, and Body Weight of Mice

Plasma PSA levels were higher in all tumor-bearing mice (*n* = 12) compared with non-tumor-bearing mice (*n* = 10), which had PSA levels below 0.003 ng/mL (Table 3). In both the Selol-supplemented (Ca + Se) and placebo (Ca) groups, the plasma PSA levels at the end of the experiment were significantly higher than before treatment (*p* = 0.02771). We noticed the inverse correlation between the tumor mass and the weight of tumor-bearing mice after subtracting tumor mass (M3) at the end of the experiment (r_s_ = 0.8936, *p* = 0.0044) (Table 3), which demonstrates that as the tumor size increases, the mouse mass decreases, and the overall condition worsens.

Due to the high variability of PSA2 levels and tumor mass in the Ca + Se group, the first step involved plotting the relationship between the PSA values at the beginning (PSA1) and at the end of the study (PSA2), assuming it to be an indicator of cell number. A clear trend was observed for the Ca group, showing a positive correlation between PSA at the beginning and at the end of the experiment (R^2^ = 0.878, a = 1.846) (Figure 2a). In the Ca + Se group, it was observed that the results for two mice significantly deviated from the trend (marked with red loops) showing higher PSA levels at the end of the study than would be expected based on the initial PSA values. These mice had initial PSA (PSA1) levels greater than 30 ng/mL and were administered Se. If these individuals are excluded from the analysis, it can be noted that for mice with lower initial PSA levels (PSA1), a greater increase in PSA was observed at the end (PSA2) (R^2^ =0.626, a= −1.018) (a different trend than in the Ca group, where the relationship was directly proportional). Adding tumor mass to the analysis (Figure 2b), it can be seen that mice with small tumors in the Ca+ Se group exhibited significantly higher PSA2 levels (R^2^ = 0.7252, a = −0.0158) compared with mice with larger tumors (in the Ca group, this relationship was again directly proportional; R^2^ = 0.8686, a = 0.0057). Given that high PSA can also be a marker of cell breakdown, it can be inferred that Se is effective in the case of small tumors (PSA < 30 ng/mL at the beginning of the study) but may not be suitable as monotherapy for advanced tumors. In our case, these two individuals (PSA > 30 ng/mL) exhibited significantly larger tumor masses than predicted based on the trend line in the control group. However, the level of secreted PSA was proportional to the tumor size (estimated based on the trend line for the control group), suggesting the lack of Se action in slowing tumor growth.

We observed that in the study group (Ca + Se) with small tumors (*n =* 4), an increase in PSA2 levels and the relative increase in PSA positively correlated with higher MDA levels (B = 2.35, *p* = 0.016; B =2.21, *p* = 0.020, respectively) and negatively correlated with ORAC values (B= −2.22, *p* = 0.017; B= −1.65, *p* = 0.027, respectively). Additionally, there was a positive correlation between the animals’ weight (M3) and MDA concentration (B = 2.31, *p* = 0.0031) and a negative correlation with ORAC values (B= −2.22, *p* = 0.0030). This indicates that if we use the mice’s weight as an indicator of well-being, a higher weight is associated with increased MDA levels in tumors and decreased ORAC values. These findings appear to support the hypothesis that the induction of oxidative stress might be a mechanism through which Selol exerts its effects.

The described relationship was not observed in the placebo group (Ca). In this group, there was only a negative correlation between the relative increase in PSA and the ORAC value (B = −0.89, *p* = 0.042). In contrast, in the study group with small tumors (Ca + Se), cancer cells exhibited greater susceptibility to the pro-oxidant effects of Selol, as evidenced by the increased levels of MDA and high PSA release.

### 3.7. Histological Study

In the control group (Ca), tumor tissues from animals exhibited intact tumor cells (Figure 3a). In contrast, the Selol-treated group (Ca + Se) displayed focal necrosis ranging from 10% to 30% near blood vessels (Figure 3b). Both groups, control and Selol-treated mice, showed highly proliferative tumor cells with abnormal mitotic figures (Figure 3a–d). Furthermore, tumor tissues in the control group exhibited a compact structure (Figure 3c), whereas after Selol treatment, there was a reduction in tumor foci, with tumor cell nests surrounded by bands of fibrous stroma (Figure 3d). The sparse stroma in both groups (Ca and Ca + Se) exhibited dense vascularization without morphotic elements indicative of an inflammatory reaction.

Additionally, immunohistochemical staining for p53 and BCL2 genes, as well as the Ki-67 protein, revealed no differences in their expression between the groups (Figure 3e–j).

## 4. Discussion

This study explores the biochemical mechanism of action of Selol, a novel organic selenium compound with promising anticancer and pro-oxidant properties. It investigates how Selol induces oxidative stress, focusing on its impact on cells with heightened metabolic activity and oncogenic signaling. The production of reactive oxygen species (ROS) by pro-oxidant compounds can cause damage to cellular structures, ultimately resulting in the death of cancer cells. This discovery presents a promising opportunity for the development of therapeutic approaches in cancer treatment [27,28].

The presented study shows that Selol administration to animals with prostate cancer increased antioxidant enzyme activity and selenium concentration in tumor tissues. The ORAC value, indicating the antioxidant capacity of tumor tissues, was observed to be higher after Selol treatment. This elevated ORAC value is likely associated with the presence of selenium. It is probable that other tissues or organs in tumor-bearing mice treated with Selol would similarly exhibit enhanced ORAC values. Consistent with this, our recent studies in healthy mice have demonstrated significant increases in ORAC levels in both blood and organs following Selol supplementation [29]. However, there were no significant changes in the concentration of MDA (marker of oxidative stress) in the tumor tissues of the Selol-treated group. It is possible that the levels of oxidative stress in cancer cells may not always correlate with the level of lipid peroxidation products. The concentration of 4-hydroxynonenal, an aldehyde produced during lipid peroxidation, varies depending on the type of cancer and the stage of the tumor [29]. This variation might be due to different levels of activity of aldehyde-metabolizing enzymes that are affected by the progression of cancer, thereby impacting the concentration of aldehyde products from lipid peroxidation [30]. Selol can induce the expression of these enzymes, in the same way as antioxidant enzymes, as they share a common pathway for activation (e.g., through the Nrf2/ARE pathway, which is activated by oxidative stress) [31]. Therefore, the lack of difference in MDA levels despite ongoing oxidative stress may be justified. According to Ksiazek et al., malignant prostate cells (LNCaP) have a weaker antioxidant defense compared with normal cells (PNT1A). Therefore, tumors inducing antioxidant enzymes in response to oxidative stress might not sufficiently counteract its effects, leading to apoptotic changes in cancer cells [13]. In summary, the increase in antioxidant enzyme activity and antioxidant capacity value and the lack of differences in MDA levels in Ca-treated mice may suggest the presence of oxidative stress. This may lead to molecular changes that can result in cell death but also exert anticancer effects. To confirm this hypothesis, further investigation into other markers of oxidation is necessary.

As mentioned previously, after treatment with Selol, we observed an increase in the activity of antioxidant enzymes in tumors, specifically GSH-dependent ones, like Se-GPx, GPx, and GST. The presence of ROS (induced by Selol) causes oxidative stress, leading to the oxidation of sulfhydryl groups and a decrease in the concentration of intracellular GSH. Then, ROS activate the Nrf2-ARE pathway, which leads to the production of antioxidants and an increase in antioxidant enzyme levels in tissues. However, the heightened activity of these antioxidants, including GPx and GST, can deplete further the reduced form of GSH, which serves as a substrate for these enzymes [31]. The enzyme that reduces GSH is TrxR. Here, we observed higher activity of Se-GPx, GPx, and GST as well as antioxidant capacity in the tumor, leading to the consumption of GSH and its conversion into GSSG. However, TrxR activity remained unchanged, indicating no significant effect on the reduction of oxidized proteins that may accumulate in the cell. These findings are supported by previous research, which observed higher GSSG and lower GSH levels in the tumor after Selol treatment [15]. Further decline in glutathione concentration may result from aldehyde products formed during lipid peroxidation, reacting with glutathione (GSH) and forming rapidly eliminated conjugates [32]. All these changes lead to a decrease in the concentration of reduced GSH and an increase in the oxidative form of GSSG. This enhances the oxidoreductive potential in cancer cells, inducing them to undergo apoptosis [33,34]. Interestingly, despite administering Selol, our previous research on the same experimental group showed no significant changes in the expression of oxidative stress-related genes, such as GPx and GST, within the tumor [13]. This finding highlights the importance of phenotypic studies, including the measurement of enzymatic activities, which, although more time-consuming than gene expression screenings, provide crucial insights that might be overlooked in genetic analyses.

In evaluating the effects of Selol administration on tumor size, mouse weight (as a measure of overall well-being), and PSA levels at the end of the experiment, significant variability was observed in the treated group (CV = 62% vs. 31% for tumor weight and 70% vs. 40% for PSA2 levels). This variation suggests the presence of ongoing processes, potentially necrosis, with dynamics that vary among individuals. Based on the notable fluctuations in PSA levels and tumor sizes observed in the study, particularly after excluding individuals with large tumor masses and elevated PSA values (as shown in Section 3.6), our findings indicate that Se may be particularly effective against smaller tumors. This suggests its potential role in preventing metastasis or serving as a complementary treatment alongside therapies like chemotherapy or surgery, especially in advanced cancer stages. The obtained results support the in vitro studies on the Selol interaction with standard chemotherapy drugs, which have shown that it can significantly enhance the antiproliferative effects of doxorubicin, especially in cells resistant to the drug [11]. Additionally, in situations of vincristine-induced hyperalgesia, Selol improved the analgesic effects of fentanyl, buprenorphine, and morphine [10]. However, this hypothesis must be verified.

The elevated PSA levels observed at the end of the experiment in the small-tumor group treated with Selol could be attributed to a temporary rise in PSA, often referred to as a “PSA flare”. This phenomenon is likely caused by the breakdown of tumor cells, which releases PSA into the bloodstream as the cancer cells are destroyed [35,36]. This transient increase in PSA is not necessarily indicative of treatment failure. Instead, it reflects the dynamic process of tumor response and should be interpreted in conjunction with other clinical findings and diagnostic tests. Proper evaluation of this PSA fluctuation is essential to distinguishing between actual disease progression and a temporary flare-up due to effective treatment. Furthermore, considering the variations in oxidative stress markers, including MDA and ORAC, as well as PSA levels and animal weights, we can conclude that smaller tumors may be more sensitive to the pro-oxidant effects of Selol.

The tumor morphology in mice treated with Selol differed from that in the placebo group. These differences, supported by histopathological studies, suggest the onset of tumor cell necrosis in the Selol-treated group. Given that the tumors of animals in the Ca + Se group exhibited a different consistency, it may turn out that they are more sensitive to chemotherapy or other types of anticancer therapy. It is worth conducting research in this direction, which suggests promising new therapeutic options for palliative care, especially for patients in the final stages of cancer.

Furthermore, histological study revealed that Selol treatment influences tumors by inducing tumor cell degeneration, focal necrosis, and constriction of tumor cell fields by connective tissue. However, there was no observed effect on the degradation of the mutated p53 gene. The proliferative activity of tumor tissue remained unaffected, and there was no suppression of apoptosis, as evidenced by the absence of BCL2 oncogene expression. The mechanism of Selol’s action in this specific type of cancer may differ from its effects in other cell lines, such as A545, where both apoptosis and necrosis contribute to its mechanism [37].

In reassessing the effects of Selol on biochemical parameters, this study found no significant changes in the activities of selenium-dependent glutathione peroxidase (Se-GPx) and glutathione S-transferase (GST) in the plasma and erythrocytes of tumor-bearing mice following Selol supplementation. Additionally, the levels of MDA were comparable in both groups. However, in healthy mice, supplementation with Selol increased the activity of antioxidant enzymes in red blood cells and SeGSHPx activity and MDA concentration in plasma, which supports the previous finding that long-term Selol intake affects antioxidant enzyme activity in the blood of healthy animals [27,38]. The lack of increased antioxidant enzyme activity and the elevation in MDA levels in the plasma of tumor-bearing mice after Selol administration may be attributed to their pre-existing high levels due to the cancerous process. Tumors initiate processes like division, metabolism, and inflammatory cytokine release, leading to increased reactive oxygen species (ROS) production. This aligns with previous studies indicating heightened oxidative stress in tumor cells, impacting antioxidant enzyme activity. In various human tumors, reduced levels of superoxide dismutase and catalase activities have been observed, accompanied by an increase in GSH-dependent enzymes and thioredoxin reductase activity [37].

## 5. Conclusions

Prolonged supplementation with this novel selenium compound leads to a significant increase in the activity of antioxidant enzymes in the blood of healthy animal models, as shown previously. Selol treatment in tumor-bearing mice influences the morphology of tumor cells, inducing necrotic changes. Additionally, it affects the antioxidant–pro-oxidant potential in the tumor, likely due to oxidative stress. Se appears to increase the breakdown of cancer cells more effectively in small tumors than in larger ones. In advanced tumors, it may accelerate tumor growth if used as monotherapy. Therefore, further studies are necessary to evaluate its efficacy either in combination therapy or for the prevention of recurrence.

## Figures and Tables

**Figure 1 nutrients-16-02860-f001:**
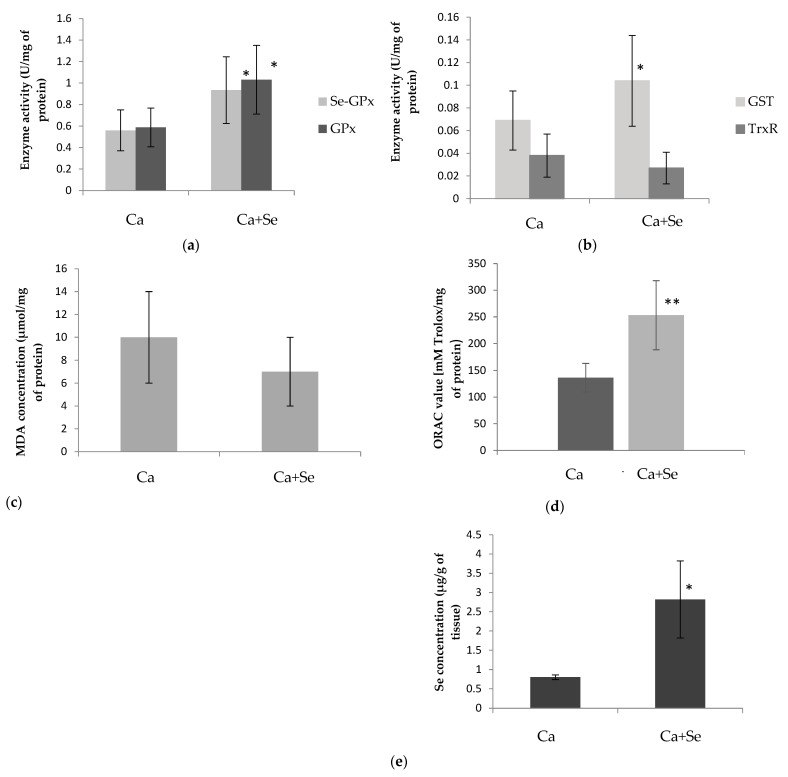
The mean enzyme activities of selenium-dependent glutathione peroxidase (Se-GPx) and total glutathione peroxidase (GPx) (**a**), and glutathione S-transferase (GST) and thioredoxin reductase (TrxR) (**b**); concentration of malondialdehyde (MDA) (**c**), ORAC value (**d**), and selenium (Se) concentration (**e**) in the tumor tissue of mice with xenografted LNCaP prostate cancer supplemented with Selol (Ca + Se) and control group of mice with xenografted LNCaP prostate supplemented with placebo (Ca). Data are shown as means ± SD (*n* = 6). The *p*-values indicate differences between control group (Ca) and study group (Ca + Se) and are indicated by * *p* < 0.05, ** *p* < 0.01.

**Figure 2 nutrients-16-02860-f002:**
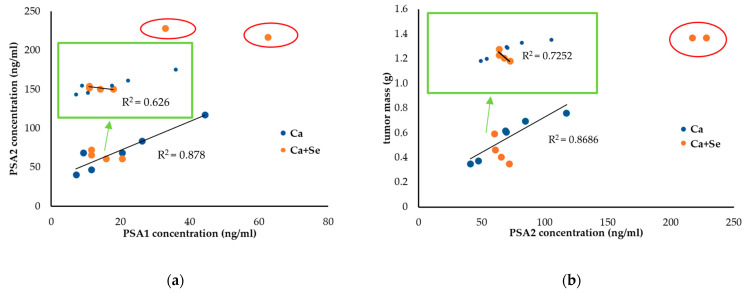
Relationship between PSA concentrations at the beginning (PSA1) and end (PSA2) of the experiment in tumor-bearing mice treated with Selol (Ca + Se, *n* = 6) or placebo (Ca, *n* = 6). Notably, two mice in the Ca + Se group significantly deviated from the expected trend (indicated by red loops) (**a**). Relationship between PSA concentration at the end of the experiment (PSA2) and tumor mass in tumor-bearing mice treated with Selol (Ca + Se, *n* = 6) or placebo (Ca, *n* = 6). Two mice in the Ca + Se group showed significant deviations from the overall trend (indicated by red loops) (**b**).

**Figure 3 nutrients-16-02860-f003:**
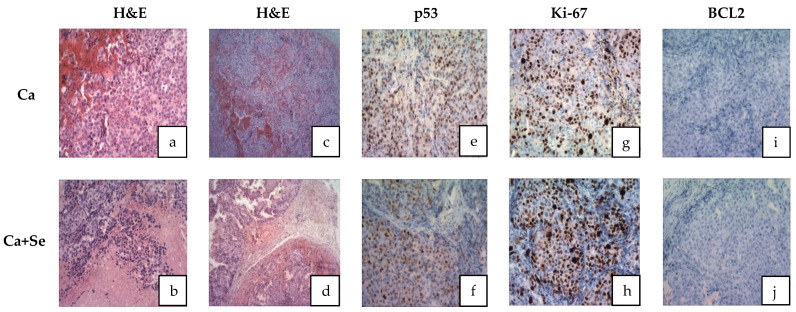
Histological photomicrographs of tumor sections stained with (**a**–**d**) H&E (hematoxylin and eosin): no features of necrosis (**a**); features of necrosis (**b**); lite tumor tissue (**c**) reduction in tumor foci (**d**); (**e**,**f**) stained with p53 (no gene expression in both cases); (**g**,**h**) stained with Ki-67 (features of high proliferation activity in both cases); and (**i**,**j**) stained with BCL2 (no gene expression in both cases). Ca + Se—mice with xenografted LNCaP prostate cancer supplemented with Selol; Ca—control group of mice with xenografted LNCaP prostate supplemented with placebo.

**Table 1 nutrients-16-02860-t001:** Selenoenzyme activities: Se-dependent (Se-GPx), total glutathione peroxidase activity (GPx), and thioredoxin activity (TrxR) in plasma and red blood cells of control mice (Control and Ca) and mice treated with Selol (Control + Se and Ca + Se). Ca + Se—mice with xenografted LNCaP prostate cancer treated with Selol; Ca—mice with xenografted LNCaP prostate cancer treated with placebo; Control—mice without prostatic tumors treated with placebo; Control + Se—mice without prostatic tumors treated with Selol.

Groups	Se-GPx Plasma (U/mL)/ RBC (U/g Hb)	GPx Plasma (U/mL)/ RBC (U/g Hb)	TrxR Plasma (U/mL)/ RBC (U/g Hb)
Ca + Se	0.83 ± 0.14 ^ab^/1.4 ± 0.5 ^a^	0.59 ± 0.12 ^a^/1.68 ± 1.38 ^c^	0.053 ± 0.008 ^a^/0.7 ± 0.5 ^b^
Ca	0.85 ± 0.11 ^ab^/1.2 ± 0.3 ^a^	0.53 ± 0.07 ^a^/1.29 ± 0.99 ^a^	0.071 ± 0.009 ^b^/1.6 ± 0.8 ^a^
Control	0.49 ± 0.07 ^a^/1.3 ± 0.2 ^a^	0.70 ± 0.13 ^b^/1.34 ± 0.75 ^a^	0.057 ± 0.011 ^a^/1.5 ± 0.7 ^a^
Control + Se	0.63 ± 0.07 ^b^/2.3 ± 0.4 ^b^	0.82 ± 0.19 ^b^/2.22 ± 1.02 ^b^	0.052 ± 0.009 ^a^/1.4 ± 0.2 ^a^

Data are presented as means ± standard deviation (SD). For the control groups (Control and Control + Se), *n* = 5, and for the tumor-bearing groups (Ca and Ca + Se), *n* = 6. The mean values in columns with different superscript letter(s) are significantly different at *p* < 0.05, as assessed by Tukey’s post hoc test.

**Table 2 nutrients-16-02860-t002:** Glutathione S-transferase activity (GST), malondialdehyde concentration (MDA), and selenium concentration (Se) in plasma and/or red blood cells of control mice (Control and Ca) and mice treated with Selol (Control + Se and Ca + Se). Ca + Se—mice with xenografted LNCaP prostate cancer treated with Selol; Ca—mice with xenografted LNCaP prostate cancer treated with placebo; Control—mice without prostatic tumors treated with placebo; Control + Se—mice without prostatic tumors treated with Selol.

Groups	GST Plasma (U/mL)/ RBC (U/g Hb)	MDA Plasma (nmol/mL)	Se RBC (µg/g Hb)
Ca + Se	0.063 ± 0.004 ^b^/0.064 ± 0.001 ^c^	78 ± 19 ^b^	1.00 ± 0.34 ^d^
Ca	0.064 ± 0.005 ^b^/0.062 ± 0.005 ^c^	87 ± 15 ^b^	0.54 ± 0.14 ^c^
Control	0.041 ± 0.001 ^a^/0.045 ± 0.001 ^a^	72 ± 12 ^a^	0.40 ± 0.09 ^a^
Control + Se	0.046 ± 0.002 ^a^/0.054 ± 0.001 ^b^	86 ± 9 ^b^	0.76 ± 0.08 ^b^

Data are presented as means ± standard deviation (SD). For the control groups (Control and Control + Se), *n* = 5, and for the tumor-bearing groups (Ca and Ca + Se), *n* = 6. The mean values in columns with different superscript letter(s) are significantly different at *p* < 0.05, as assessed by Tukey’s post hoc test.

**Table 3 nutrients-16-02860-t003:** Tumor mass, plasma prostate-specific antigen (PSA) concentration before (PSA1) and after (PSA2) Selol/placebo administration, and weight of mice minus weight of isolated tumor at the end of experiment (M3). Ca + Se—mice with xenografted LNCaP prostate cancer treated with Selol; Ca—mice with xenografted LNCaP prostate cancer treated with placebo; Control—mice without prostatic tumors treated with placebo; Control + Se—mice without prostatic tumors treated with Selol.

Groups	Tumor Mass (g)	PSA1 (ng/mL)	PSA2 (ng/mL)	M3 (g)
Ca + Se	0.75 ± 0.48 ^a^	26 ± 19 ^a^	117 ± 81 ^a^	26.9 ± 3.5 ^a^
Ca	0.56 ± 0.17 ^a^	20 ± 14 ^a^	71 ± 28 ^a^	26.57 ± 0.92 ^a^
Control	-	<0.003	<0.003	27.1 ± 1.3 ^a^
Control + Se	-	<0.003	<0.003	28.4 ± 0.9 ^a^

Data are presented as means ± standard deviation (SD). For the control groups (Control and Control + Se), *n* = 5, and for the tumor-bearing groups (Ca and Ca + Se), *n* = 6. The mean values in columns with different superscript letter(s) are significantly different at *p* < 0.05, as assessed by Tukey’s post hoc test.

## Data Availability

The data presented in this study are available upon request form corresponding author.

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
