# Peer review of "Effect of Selol on Tumor Morphology and Biochemical Parameters Associated with Oxidative Stress in a Prostate Tumor-Bearing Mice Model"

_nutrients, 2024, doi:10.3390/nu16172860_

Round 1

Reviewer 1 Report

Comments and Suggestions for Authors

This manuscript by Małgorzata Sochacka et al. investigates the effectiveness of Selol treatment in a mouse model of prostate cancer using LNCaP cells. The study demonstrates, as expected, that Selol treatment increased the activity of antioxidant enzymes (Se-GPx, GPx, and GST) in tumors compared to control groups.

However, a key limitation of the study is the lack of investigation into the relationship between the observed enzyme activity changes and actual tumor growth inhibition.  The manuscript does not address whether Selol treatment resulted in a decrease in tumor burden.  Furthermore, it's unclear if the measured enzyme activity can be used as a predictive marker for treatment efficacy.

In addition, the author's choice of MDA concentration to assess oxidative stress appears to have yielded inconclusive results between the control and experimental groups. While acknowledging the complexity of the system, evaluating the efficacy of an antioxidant drug requires a more robust approach to quantify relevant biomarkers.

This manuscript primarily offers qualitative observations on the treatment's performance, illustrated in histology, lacking the quantitative data necessary to definitively assess its effectiveness. To strengthen the conclusions and meet publication standards, the authors should consider including: 1) measurement of specific ROS species that is relevant to Selol's mechanism or relevant oxidative stress biomarker; 2) connect enzyme activity with tumor growth data.

Author Response

Dear Reviewer,

We would like to thank you for the time and effort taken to review our manuscript. All comments and remarks were carefully considered and addressed. Your suggestions led us to include missing data, which may be crucial for a better understanding of the effects of Selol on prostate tumors. We hope that the implemented corrections, which have enhanced the quality of our manuscript, will be satisfactory, and that our manuscript will be considered for publication. Please find below the responses.

Comment 1: This manuscript by Małgorzata Sochacka et al. investigates the effectiveness of Selol treatment in a mouse model of prostate cancer using LNCaP cells. The study demonstrates, as expected, that Selol treatment increased the activity of antioxidant enzymes (Se-GPx, GPx, and GST) in tumors compared to control groups.

However, a key limitation of the study is the lack of investigation into the relationship between the observed enzyme activity changes and actual tumor growth inhibition.  The manuscript does not address whether Selol treatment resulted in a decrease in tumor burden. 

Response 1: Thank you for that comment. We have added the data on the tumor growth and prostate-specific antigen (PSA) level, a marker primarily used for screening and monitoring prostate cancer. We have obtained very interesting observations.

3.5 Morphological study

The tumors of both the study and placebo mice displayed a range of sizes, all characterized by noticeable blood vessel proliferation. Significant differences in tumor appearance were observed between the two experimental groups. Tumors in mice treated with Selol (Ca+Se) appeared darker in color, exhibited swollen interiors, and in some regions, had a "jelly-like" consistency. In contrast, tumors in mice treated with the placebo (Ca) maintained a dense and firm structure. Furthermore, there were no observable alterations in the tissue appearance of other organs in mice that received Selol (Ca+Se, Control+Se).

3.6 PSA concentration, tumor mass, and body weight of mice

Plasma PSA levels were higher in all tumor-bearing mice (n=10) compared to non-tumor-bearing mice (n=10), which had PSA levels below 0.003 ng/ml (Table 3). In both the Selol-supplemented (Ca+Se) and placebo (Ca) groups, plasma PSA levels at the end of the experiment were significantly higher than before treatment (p=0.02771). We did not observe the significant differences between the PSA2 levels, at the end of the experiment in the group with tumors supplemented with Selol or placebo (Ca+Se vs Ca). Furthermore, there was a statistically significant correlation between PSA concentration at the end of the experiment (PSA2) and tumor mass in the placebo group (Ca) (rs=0.9429, p=0.0048), with no correlation in the study group (Ca+Se). At the end of the experiment, there were no significant differences in mean tumor masses or body weight between the Selol-supplemented (Ca+Se) and placebo (Ca) groups, or between the control (Ca) and Selol-treated (Control+Se, Ca+Se) groups (Table 3). We noticed the inverse correlation between the tumor mass and the weight of tumor-bearing mice after subtracting tumor mass (M3) at the end of experiment (rs=0.8936, p=0.0044) (Table 3), which demonstrates that as the tumor size increases, the mouse mass decreases and the overall condition worsens.”

Comment 2: Furthermore, it’s unclear if the measured enzyme activity can be used as a predictive marker for treatment efficacy.

Response 2: We have only observed the increase in the activity of antioxidant enzymes in tumors of mice treated with Se, specifically GSH dependent like Se-GPx, GPx, and GST. So, at the moment there is no evidence that the enzyme activity level can be a marker of treatment efficiency.

Comment 3: In addition, the author's choice of MDA concentration to assess oxidative stress appears to have yielded inconclusive results between the control and experimental groups. While acknowledging the complexity of the system, evaluating the efficacy of an antioxidant drug requires a more robust approach to quantify relevant biomarkers.

Response 3: We agree with the Reviewer that many compounds allow us to evaluate the level of oxidative stress in the organism. These can include oxidatively modified lipids, proteins, amino acids, etc. We selected MDA because this marker was strongly affected by Selol in all previous analyses on healthy mice (in tissues and plasma). We have added this information as a limitation of our study

“Therefore, the lack of difference in MDA levels despite ongoing oxidative stress may be justified ”

“In summary, the increase in antioxidant enzyme activity and antioxidant capacity value and the lack of differences in MDA levels in Ca-treated mice suggest the presence of oxidative stress”

Moreover, based on the comment we have added the data for ORAC, one of the most accepted and accurate indicators of antioxidant status.

“ORAC value

The tumor tissue oxygen-radical absorbance capacity (ORAC) was significantly higher in the group of mice treated with Selol (Ca+Se) compared to the placebo group (Ca) (Figure 1d). No significant correlations were found between the ORAC value of tumors and other measured tumor parameters”

Comment 4: This manuscript primarily offers qualitative observations on the treatment's performance, illustrated in histology, lacking the quantitative data necessary to definitively assess its effectiveness. To strengthen the conclusions and meet publication standards, the authors should consider including 1) measurement of specific ROS species that is relevant to Selol's mechanism or relevant oxidative stress biomarker; 2) connection enzyme activity with tumor growth data.

Response 4: Actually, the mechanism of Selol's action is not fully understood. We do not know which ROS species should be measured. Moreover, as we can see, different oxidative statuses are observed in plasma and tumors. Since ROS are short-lived molecules, we have no idea how to obtain reliable ROS levels in the tumor. However, as the Reviewer suggested, we added some data on tumor growth, mice weight, PSA level, and ORAC value, the indicator of total radical scavenging ability.

Reviewer 2 Report

Comments and Suggestions for Authors

Authors aimed to investigate effects of Selol, a selenintriglyceride compound derived from sunflower oil with potential anticancer and pro-oxidant properties, on antioxidant enzyme activities (Se-GPx, GPx, GST, and TrxR), plasma malondialdehyde (MDA) concentration, selenium (Se) concentration, tumor morphology and the expression of p53, BCL2, and Ki-67 in mice with xenografted LNCaP prostate cancer. Treatment included 5% Selol at a dose of 17 mg Se per kg of body mass and results were compared between four groups of animals (N=5): Ca - mice with xenografted LNCaP prostate cancer, Control - mice without prostatic tumors, Control+Se - mice without prostatic tumors treated with Selol, Ca+Se - mice with xenografted LNCaP prostate cancer treated with Selol. Treatment resulted with necrotic changes, increased antioxidant enzyme activities (Se-GPx, total GPx, and GST) and selenium concentration in blood and tumor tissues. This topic could be interesting and important, however there are several key points that need to be addressed.

L117/118: What is “appropriate” anesthesia?

L 137, “(dose based on the in vivo results from the Selol toxicity study; unpublished data)”. More details should be given – how the specified dose of Selol (5% at 17 mg/kg body mass) was selected.

L 161 and further: The authors state that enzymatic analyses were performed in red blood cells, but in L 149-152 it says that antioxidant enzymes were measured in the supernatant of the thawed and hemolyzed “blood” i.e.  “red blood cells”. This should be corrected.

It is not clear what is determined in which matrix. According to the L 161, Se-GPx, total GPx, TrxR, GST, MDA, and Se concentration were determined “in the plasma, red blood cells, and supernatants of the tumor homogenates”. Therefore, readers will expect to see the methods described and the results of all these parameters in all three matrices. It should be specified which parameter is measured in which matrix and appropriate methods for each combination parameter/matrix should be provided. When parameter was measured in the supernatant of the hemolyzed RBCs and/or in the supernatant of the homogenized tumor tissues, this should be clearly stated.

Section 2.4.4., L209, “The erythrocytes and tissue samples were mineralized”, L 210/211, “The optimal measuring range for the ICP-MS method for this element was 0.1-2.5 μg/L”, and L 211/212, “Se concentration was expressed as μg/g of tissue”. If the Se was measured in erythrocytes/supernatant of the hemolyzed erythrocytes, results should be expressed per g of hemoglobin. Why “the optimal Se concentration” in μg/L was mentioned? Is this concentration of Se for healthy humans in serum, plasma, or whole blood? Or this is a dynamic range for Se measured by ICP-MS in one of these matrices?  Is any quality control sample with the reference range for the Se used in the study to check accuracy of the measurements? Moreover, in the cited reference 26, AAS method for Se in serum and urine is described.

L 224, “Antibodies for p53, BCL2 and Ki-67 were used”: Please, specify.

The results should be presented in such a way that it is obvious which parameter was measured in which matrix. Accordingly, the appropriate units of measurement should be harmonized. For example, for malondialdehyde, section 2.4.2., there is no information in which matrix is MDA measured, but L 196 states that “MDA concentration was expressed as μmol/mg of protein”. However, results presented in table 2 are for plasma and are expressed as nmol/ml. Similar, for TrxR activity, L183-192, only method for supernatant is described with “The results were expressed as U/mg protein”. However, in the Table 2, the results are presented for plasma and RBCs. There is no method description for MDA and TrxR in plasma. Please, check these data for all parameters and correct accordingly.

The tables and figures should be clear and understandable without reading the text of the manuscript. It should be clear that there are mice with xenografted LNCaP prostate cancer, mice without prostatic tumors, mice without prostatic tumors treated with Selol, and mice with xenografted LNCaP prostate cancer treated with Selol.

Section 3.3.: Title of this section is “Selenium concentrations in erythrocytes”, but the results for Se in tumors are also presented.

 L 302303, “This section may be divided by subheadings. It should provide a concise and precise description of the experimental results, their interpretation, as well as the experimental conclusions that can be drawn” should be deleted.

Author Response

Dear Reviewer,

Sincerest thank you for taking the time to review this manuscript: “Effect of Selol on Tumor Morphology and Biochemical Parameters Associated with Oxidative Stress in a Prostate Tumor-Bearing Mice Model”. We have modified the paper in response to the extensive and insightful Reviewer comments.

Comments 1: L117/118: What is “appropriate” anesthesia?

Response 1: Thank you for pointing this out. We agree with this comment. Therefore, we have deleted this text, and change it to “At the end of the experiment, mice were anesthetized with halothane…. “ mentioned later in the Experimental protocol Section (page 3; Section 2.3; L145/146).

Comments 2: L137, “(dose based on the in vivo results from the Selol toxicity study; unpublished data)”. More details should be given – how the specified dose of Selol (5% at 17 mg/kg body mass) was selected.

Response 2: We have added the explanation “(which is approx. 20% of LD50 - dose based on the in vivo results from the Selol toxicity study; unpublished data) add in the text: Experimental protocol Section (page 3; Section 2.3; L140/141).

Comment 3: L 161 and further: The authors state that enzymatic analyses were performed in red blood cells, but in L 149-152 it says that antioxidant enzymes were measured in the supernatant of the thawed and hemolyzed “blood” i.e.  “red blood cells”. This should be corrected.

Response 3: The appropriate correction was made and changed “blood” to “red blood cells/erythrocytes” word: Biochemical analysis Section (page 4; Section 2.4; L155).

Comment 4: It is not clear what is determined in which matrix. According to the L 161, Se-GPx, total GPx, TrxR, GST, MDA, and Se concentration were determined “in the plasma, red blood cells, and supernatants of the tumor homogenates”. Therefore, readers will expect to see the methods described and the results of all these parameters in all three matrices. It should be specified which parameter is measured in which matrix and appropriate methods for each combination parameter/matrix should be provided. When parameter was measured in the supernatant of the hemolyzed RBCs and/or in the supernatant of the homogenized tumor tissues, this should be clearly stated.

Response 4: Thank you for pointing this out. We changed and clearly emphasized in which matrix the determined parameters were measured: Biochemical analysis Section (page 4; Section 2.4; L168/174): “The following parameters were determined in plasma, erythrocytes and tumor homogenate supernatants: selenium-dependent glutathione peroxidase (Se-GPx) and total glutathione peroxidase (GPx) activities, thioredoxin reductase (TrxR) and glutathione S-transferase (GST) activities. Additionally, plasma PSA levels, plasma and tumor malondialdehyde (MDA, lipid peroxidation marker) concentration and erythrocyte and tumor Se concentrations were measured”.

Comment 5: Section 2.4.4., L209, “The erythrocytes and tissue samples were mineralized”, L 210/211, “The optimal measuring range for the ICP-MS method for this element was 0.1-2.5 μg/L”, and L 211/212, “Se concentration was expressed as μg/g of tissue”. If the Se was measured in erythrocytes/supernatant of the hemolyzed erythrocytes, results should be expressed per g of hemoglobin. Why “the optimal Se concentration” in μg/L was mentioned? Is this concentration of Se for healthy humans in serum, plasma, or whole blood? Or this is a dynamic range for Se measured by ICP-MS in one of these matrices?  Is any quality control sample with the reference range for the Se used in the study to check accuracy of the measurements? Moreover, in the cited reference 26, AAS method for Se in serum and urine is described.

Response 5:  The appropriate correction was made. The correct units were given in the Table 2: Se RBC (µg/g Hb) and in the Figure 1: Se concentration (µg/g of tissue): (page 7; Section 3.2, Section 3.4).

We added the Reference 27: Forrer R, Gautschi K, Stroh A, Lutz H. Direct determination of selenium and other trace elements in serum samples by ICP-MS. J Trace Elem Med Biol. 1999 Apr;12(4):240-7.

Comment 6: L224, “Antibodies for p53, BCL2 and Ki-67 were used”: Please, specify.

Response 6:  Thank you for that comment. We have added the missing information: Histological study section (page 6; Section 2.5; L254/255): “Antibodies against p53, BCL2 and Ki-67 (DAKO Omnis, Agilent) were used to assess protein expression levels and to evaluate tumor characteristics”.

Comment 7: The results should be presented in such a way that it is obvious which parameter was measured in which matrix. Accordingly, the appropriate units of measurement should be harmonized. For example, for malondialdehyde, section 2.4.2., there is no information in which matrix is MDA measured, but L 196 states that “MDA concentration was expressed as μmol/mg of protein”. However, results presented in table 2 are for plasma and are expressed as nmol/ml. Similar, for TrxR activity, L183-192, only method for supernatant is described with “The results were expressed as U/mg protein”. However, in the Table 2, the results are presented for plasma and RBCs. There is no method description for MDA and TrxR in plasma. Please, check these data for all parameters and correct accordingly.

Response 7: Thank you for pointing this out. We have modified the description of methods and have removed the data related to the units in which the measured parameters are given. Presented methods are the same for tissue supernatant and plasma. To harmonized the data, units were given in the presentation of particular values in tables and figures. We add the information in which matrix the parameter was measured.

Comment 8: The tables and figures should be clear and understandable without reading the text of the manuscript. It should be clear that there are mice with xenografted LNCaP prostate cancer, mice without prostatic tumors, mice without prostatic tumors treated with Selol, and mice with xenografted LNCaP prostate cancer treated with Selol.

Response 8:  Thank you for that comment. We have made the corrections and modified the description of Tabels (Table 1 and Table 2) and Figure 1.

Comment 9: Section 3.3.: Title of this section is “Selenium concentrations in erythrocytes”, but the results for Se in tumors are also presented.

Response 9: We have renamed the section for: “Selenium concentration”:

Comment 10: L 302303, “This section may be divided by subheadings. It should provide a concise and precise description of the experimental results, their interpretation, as well as the experimental conclusions that can be drawn” should be deleted.

Response 10:  Thank you for pointing out our oversight.

Round 2

Reviewer 1 Report

Comments and Suggestions for Authors

In this updated manuscript, the authors have introduced two new sets of results: a PSA level study and tumor ORAC measurement.

For the PSA level study, the data presented in the treatment group, including tumor mass, PSA levels at the end of the study, and mouse weight, showed significant variation compared to the control group. This variability makes it difficult to draw conclusions about whether Selol treatment promotes or inhibits tumor growth. Although the mean values of tumor mass and PSA levels increased in the treatment group compared to the control group—typically indicative of tumor promotion—I agree with the authors’ argument that this does not necessarily mean the treatment is failing. However, this data does not convincingly demonstrate the treatment’s success either.

Regarding the ORAC measurement, I agree with the authors’ argument that the increased ORAC value is attributable to the presence of selenium in the tumor tissues. It is likely that other tissues or organs from mice treated with Selol would also show increased ORAC values.

The authors need to provide quantitative measurements to convincingly demonstrate Selol’s effect on inhibiting tumor growth in vivo. Such biochemical parameter measurements will enhance our understanding of the mechanisms behind Selol treatment and facilitate drug development.

While selenium’s anti-cancer properties are well-known, its toxicity to healthy cells remains a significant challenge. The key is to develop a drug that can selectively target tumor cells. Therefore, the histological photomicrographs of necrosis (tumor morphology) in the treatment group alone are not convincing evidence to prove Selol’s superiority over other reported selenium compounds.

Author Response

Dear Reviewer, 

Thank you for your thorough review. We have carefully addressed all your comments and incorporated the necessary data to enhance the understanding of Selol's effects on prostate tumors. We believe these revisions have strengthened our manuscript and hope it meets your approval. Please find our detailed responses below.

Comment 1: In this updated manuscript, the authors have introduced two new sets of results: a PSA level study and tumor ORAC measurement.

For the PSA level study, the data presented in the treatment group, including tumor mass, PSA levels at the end of the study, and mouse weight, showed significant variation compared to the control group. This variability makes it difficult to draw conclusions about whether Selol treatment promotes or inhibits tumor growth. Although the mean values of tumor mass and PSA levels increased in the treatment group compared to the control group—typically indicative of tumor promotion—I agree with the authors’ argument that this does not necessarily mean the treatment is failing. However, this data does not convincingly demonstrate the treatment’s success either.

The authors need to provide quantitative measurements to convincingly demonstrate Selol’s effect on inhibiting tumor growth in vivo. Such biochemical parameter measurements will enhance our understanding of the mechanisms behind Selol treatment and facilitate drug development.

While selenium’s anti-cancer properties are well-known, its toxicity to healthy cells remains a significant challenge. The key is to develop a drug that can selectively target tumor cells. Therefore, the histological photomicrographs of necrosis (tumor morphology) in the treatment group alone are not convincing evidence to prove Selol’s superiority over other reported selenium compounds.

Response 1: We appreciate the valuable feedback provided. We acknowledge the variability in the data from the PSA level study, including tumor mass, and mouse weight, which indeed presents challenges in drawing definitive conclusions regarding the effects of Selol treatment. While the increase in mean tumor mass and PSA levels in the treatment group could suggest tumor promotion, it is important to note that this outcome may also result from other factors, such as tumor cell breakdown or variations in tumor response to treatment. We agree that this data alone does not conclusively demonstrate the success of the treatment. The Reviewer's comment prompted us to take a closer look at the obtained results. Therefore, we conducted additional statistical analyses, we have add some additional information to: Results, Paragraph 3.6: PSA concentration, tumor mass, and body weight of mice:

“Due to the high variability of PSA2 levels and tumor mass in the group Ca+Se, the first step involved plotting the relationship between PSA values at the beginning (PSA1) and the end of the study (PSA2), assuming it to be an indicator of cell number. A clear trend was observed for the Ca group, showing a positive correlation between PSA at the beginning and the end of the experiment (R2=0.878, a=1.846) (Figure 2a). In the Ca+Se group, it was observed that the results for two mice significantly deviated from the trend (marked with red loops) showing higher PSA levels at the end of the study than would be expected based on the initial PSA values. These mice had initial PSA (PSA1) levels greater than 30 ng/ml and were administered Se. If these individuals are excluded from the analysis, it can be noted that for mice with lower initial PSA levels (PSA1), a greater increase in PSA was observed at the end (PSA2) (R2 =0.640, a= -1.018) (a different trend than in the Ca group, where the relationship was direct proportional). Adding tumor mass to the analysis (Figure 2b), it can be seen that mice with small tumors in Ca+ Se group exhibited significantly higher PSA2 levels (R2=0.7138, a=-0.0158) compared to mice with larger tumors (in the Ca group, this relationship was again direct proportional; R2=0.8686, a=0.0057). Given that high PSA can also be a marker of cell breakdown, it can be inferred that Se is effective in the case of small tumors (PSA< 30 ng/ml at the beginning of the study), but may not be suitable as monotherapy for advanced tumors. In our case, these two individuals (PSA>30 ng/ml) exhibited significantly larger tumor masses than predicted based on the trend line in the control group. However, the level of secreted PSA was proportional to the tumor size (estimated based on the trend line for the control group), suggesting the lack of Se action in slowing tumor growth.”

Discussion section: In evaluating the effects of Selol administration on tumor size, mouse weight (as a measure of overall well-being), and PSA levels at the end of the experiment, significant variability was observed in the treated group (CV=62% vs. 31% for tumor weight and 70% vs. 40% for PSA2 levels). This variation suggests the presence of ongoing processes, potentially necrosis, with dynamics that vary between individuals. Given the significant fluctuations in PSA levels and tumor size observed in the studied group of mice, and excluding those individuals characterized by large tumor masses and high PSA values (Results 3.6), our results suggest that Se may be effective against small tumors, potentially proving its efficacy in preventing metastasis or as an adjunctive treatment following therapies such as chemotherapy, or surgery, particularly in advanced stages of cancer. The obtained results support the in vitro studies on the Selol interaction with standard chemotherapy drugs, shown it can significantly enhance the antiproliferative effects of Doxorubicin, especially in cells resistant to the drug [11]. Additionally, in situations of vincristine-induced hyperalgesia, Selol improved the analgesic effects of fentanyl, buprenorphine, and morphine [10]. However, this hypothesis must be verified.

The elevated PSA levels observed at the end of the experiment in the small tumor group treated with Selol could be attributed to a temporary rise in PSA, often referred to as a "PSA flare." This phenomenon is likely caused by the breakdown of tumor cells, which releases PSA into the bloodstream as the cancer cells are destroyed [36, 37]. This transient increase in PSA is not necessarily indicative of treatment failure. Instead, it reflects the dynamic process of tumor response and should be interpreted in conjunction with other clinical findings and diagnostic tests. Proper evaluation of this PSA fluctuation is essential to distinguish between actual disease progression and a temporary flare-up due to effective treatment.

Furthermore, considering the variations in oxidative stress markers, including MDA and ORAC, as well as PSA levels and animal weights, we can conclude that smaller tumors may be more sensitive to the pro-oxidant effects of Selol.

Tumor morphology in mice treated with Selol differed from that in the placebo group. These differences, supported by histopathological studies, suggest the onset of tumor cell necrosis in the Selol-treated group. Given that the tumors of animals in the Ca+Se group exhibited a different consistency, it may turn out that they are more sensitive to chemotherapy or other types of anticancer therapy. It is worth conducting research in this direction and suggest promising new therapeutic options for palliative care, especially for patients in the final stages of cancer.

And in the Conclusions section: “Se appears to increase the breakdown of cancer cells more effectively in small tumors than in larger ones. In advanced tumors, it may accelerate tumor growth if used as monotherapy. Therefore, further studies are necessary to evaluate its efficacy either in combination therapy or for the prevention of recurrence.”

This is a crucial result because in Poland, Selol is sometimes administered to patients when no other treatment options are available. The significance of this finding lies in the potential of Selol as a last-resort therapy, offering hope for patients who have exhausted conventional treatments. Selol is used as a supportive therapy alongside standard treatments such as chemotherapy, radiation therapy, or surgery, particularly in advanced stages of cancer. The goal is to enhance the overall effectiveness of the primary treatment, potentially improving outcomes or reducing side effects. In this context, Selol's role is to complement conventional therapies rather than replace them, helping to manage the disease more effectively. Understanding its effectiveness and safety profile is essential for considering its broader application in oncology.

The efficacy of Selol's action was proven in in vitro studies. Therefore, our research is focused on elucidating in vivo the biochemical pathways through which Selol exerts its effects. While the efficacy of Selol is undoubtedly important, our primary goal is to uncover how it interacts with cellular processes, especially at the molecular and enzymatic levels. That’s why we add to the Result section: We observed that in the study group (Ca+Se) with small tumors (n=4), an increase in PSA2 levels and the relative increase in PSA positively correlated with higher MDA levels (B=2.35, p=0.016; B =2.21, p=0.020 respectively) and negatively correlated with ORAC values (B= -2.22, p=0.017; B= -1.65, p=0.027 respectively). Additionally, there was a positive correlation between the animals' weight (M3) and MDA concentration (B=2.31, p=0.0031), and a negative correlation with ORAC values (B= -2.22, p=0.0030). This indicates that if we use the mice's weight as an indicator of well-being, a higher weight is associated with increased MDA levels in tumors and decreased ORAC values. These findings appear to support the hypothesis that the induction of oxidative stress might be a mechanism through which Selol exerts its effects.

The described relationship was not observed in the placebo group (Ca). In this group, there was only a negative correlation between the relative increase in PSA and the ORAC value (B = -0.89, p=0.042). In contrast, in the study group, with small tumors (Ca+Se), cancer cells exhibited greater susceptibility to the pro-oxidant effects of Selol, as evidenced by the increased levels of MDA and high PSA release.

This approach helps us contribute to the broader understanding of how Selol functions, potentially leading to more targeted and effective therapeutic applications. Our previous studies -Reference [15, 25, 27] have provided insights into these mechanisms, and we aim to build upon this foundation to deepen the scientific community’s knowledge of Selol’s action.

Comment 1: In this updated manuscript, the authors have introduced two new sets of results: a PSA level study and tumor ORAC measurement.

For the PSA level study, the data presented in the treatment group, including tumor mass, PSA levels at the end of the study, and mouse weight, showed significant variation compared to the control group. This variability makes it difficult to draw conclusions about whether Selol treatment promotes or inhibits tumor growth. Although the mean values of tumor mass and PSA levels increased in the treatment group compared to the control group—typically indicative of tumor promotion—I agree with the authors’ argument that this does not necessarily mean the treatment is failing. However, this data does not convincingly demonstrate the treatment’s success either.

The authors need to provide quantitative measurements to convincingly demonstrate Selol’s effect on inhibiting tumor growth in vivo. Such biochemical parameter measurements will enhance our understanding of the mechanisms behind Selol treatment and facilitate drug development.

While selenium’s anti-cancer properties are well-known, its toxicity to healthy cells remains a significant challenge. The key is to develop a drug that can selectively target tumor cells. Therefore, the histological photomicrographs of necrosis (tumor morphology) in the treatment group alone are not convincing evidence to prove Selol’s superiority over other reported selenium compounds.

Response 1: We appreciate the valuable feedback provided. We acknowledge the variability in the data from the PSA level study, including tumor mass, and mouse weight, which indeed presents challenges in drawing definitive conclusions regarding the effects of Selol treatment. While the increase in mean tumor mass and PSA levels in the treatment group could suggest tumor promotion, it is important to note that this outcome may also result from other factors, such as tumor cell breakdown or variations in tumor response to treatment. We agree that this data alone does not conclusively demonstrate the success of the treatment. The Reviewer's comment prompted us to take a closer look at the obtained results. Therefore, we conducted additional statistical analyses, we have add some additional information to: Results, Paragraph 3.6: PSA concentration, tumor mass, and body weight of mice:

“Due to the high variability of PSA2 levels and tumor mass in the group Ca+Se, the first step involved plotting the relationship between PSA values at the beginning (PSA1) and the end of the study (PSA2), assuming it to be an indicator of cell number. A clear trend was observed for the Ca group, showing a positive correlation between PSA at the beginning and the end of the experiment (R2=0.878, a=1.846) (Figure 2a). In the Ca+Se group, it was observed that the results for two mice significantly deviated from the trend (marked with red loops) showing higher PSA levels at the end of the study than would be expected based on the initial PSA values. These mice had initial PSA (PSA1) levels greater than 30 ng/ml and were administered Se. If these individuals are excluded from the analysis, it can be noted that for mice with lower initial PSA levels (PSA1), a greater increase in PSA was observed at the end (PSA2) (R2 =0.640, a= -1.018) (a different trend than in the Ca group, where the relationship was direct proportional). Adding tumor mass to the analysis (Figure 2b), it can be seen that mice with small tumors in Ca+ Se group exhibited significantly higher PSA2 levels (R2=0.7138, a=-0.0158) compared to mice with larger tumors (in the Ca group, this relationship was again direct proportional; R2=0.8686, a=0.0057). Given that high PSA can also be a marker of cell breakdown, it can be inferred that Se is effective in the case of small tumors (PSA< 30 ng/ml at the beginning of the study), but may not be suitable as monotherapy for advanced tumors. In our case, these two individuals (PSA>30 ng/ml) exhibited significantly larger tumor masses than predicted based on the trend line in the control group. However, the level of secreted PSA was proportional to the tumor size (estimated based on the trend line for the control group), suggesting the lack of Se action in slowing tumor growth.”

Discussion section: In evaluating the effects of Selol administration on tumor size, mouse weight (as a measure of overall well-being), and PSA levels at the end of the experiment, significant variability was observed in the treated group (CV=62% vs. 31% for tumor weight and 70% vs. 40% for PSA2 levels). This variation suggests the presence of ongoing processes, potentially necrosis, with dynamics that vary between individuals. Given the significant fluctuations in PSA levels and tumor size observed in the studied group of mice, and excluding those individuals characterized by large tumor masses and high PSA values (Results 3.6), our results suggest that Se may be effective against small tumors, potentially proving its efficacy in preventing metastasis or as an adjunctive treatment following therapies such as chemotherapy, or surgery, particularly in advanced stages of cancer. The obtained results support the in vitro studies on the Selol interaction with standard chemotherapy drugs, shown it can significantly enhance the antiproliferative effects of Doxorubicin, especially in cells resistant to the drug [11]. Additionally, in situations of vincristine-induced hyperalgesia, Selol improved the analgesic effects of fentanyl, buprenorphine, and morphine [10]. However, this hypothesis must be verified.

The elevated PSA levels observed at the end of the experiment in the small tumor group treated with Selol could be attributed to a temporary rise in PSA, often referred to as a "PSA flare." This phenomenon is likely caused by the breakdown of tumor cells, which releases PSA into the bloodstream as the cancer cells are destroyed [36, 37]. This transient increase in PSA is not necessarily indicative of treatment failure. Instead, it reflects the dynamic process of tumor response and should be interpreted in conjunction with other clinical findings and diagnostic tests. Proper evaluation of this PSA fluctuation is essential to distinguish between actual disease progression and a temporary flare-up due to effective treatment.

Furthermore, considering the variations in oxidative stress markers, including MDA and ORAC, as well as PSA levels and animal weights, we can conclude that smaller tumors may be more sensitive to the pro-oxidant effects of Selol.

Tumor morphology in mice treated with Selol differed from that in the placebo group. These differences, supported by histopathological studies, suggest the onset of tumor cell necrosis in the Selol-treated group. Given that the tumors of animals in the Ca+Se group exhibited a different consistency, it may turn out that they are more sensitive to chemotherapy or other types of anticancer therapy. It is worth conducting research in this direction and suggest promising new therapeutic options for palliative care, especially for patients in the final stages of cancer.

And in the Conclusions section: “Se appears to increase the breakdown of cancer cells more effectively in small tumors than in larger ones. In advanced tumors, it may accelerate tumor growth if used as monotherapy. Therefore, further studies are necessary to evaluate its efficacy either in combination therapy or for the prevention of recurrence.”

This is a crucial result because in Poland, Selol is sometimes administered to patients when no other treatment options are available. The significance of this finding lies in the potential of Selol as a last-resort therapy, offering hope for patients who have exhausted conventional treatments. Selol is used as a supportive therapy alongside standard treatments such as chemotherapy, radiation therapy, or surgery, particularly in advanced stages of cancer. The goal is to enhance the overall effectiveness of the primary treatment, potentially improving outcomes or reducing side effects. In this context, Selol's role is to complement conventional therapies rather than replace them, helping to manage the disease more effectively. Understanding its effectiveness and safety profile is essential for considering its broader application in oncology.

The efficacy of Selol's action was proven in in vitro studies. Therefore, our research is focused on elucidating in vivo the biochemical pathways through which Selol exerts its effects. While the efficacy of Selol is undoubtedly important, our primary goal is to uncover how it interacts with cellular processes, especially at the molecular and enzymatic levels. That’s why we add to the Result section: We observed that in the study group (Ca+Se) with small tumors (n=4), an increase in PSA2 levels and the relative increase in PSA positively correlated with higher MDA levels (B=2.35, p=0.016; B =2.21, p=0.020 respectively) and negatively correlated with ORAC values (B= -2.22, p=0.017; B= -1.65, p=0.027 respectively). Additionally, there was a positive correlation between the animals' weight (M3) and MDA concentration (B=2.31, p=0.0031), and a negative correlation with ORAC values (B= -2.22, p=0.0030). This indicates that if we use the mice's weight as an indicator of well-being, a higher weight is associated with increased MDA levels in tumors and decreased ORAC values. These findings appear to support the hypothesis that the induction of oxidative stress might be a mechanism through which Selol exerts its effects.

The described relationship was not observed in the placebo group (Ca). In this group, there was only a negative correlation between the relative increase in PSA and the ORAC value (B = -0.89, p=0.042). In contrast, in the study group, with small tumors (Ca+Se), cancer cells exhibited greater susceptibility to the pro-oxidant effects of Selol, as evidenced by the increased levels of MDA and high PSA release.

This approach helps us contribute to the broader understanding of how Selol functions, potentially leading to more targeted and effective therapeutic applications. Our previous studies -Reference [15, 25, 27] have provided insights into these mechanisms, and we aim to build upon this foundation to deepen the scientific community’s knowledge of Selol’s action.

Comment 2: Regarding the ORAC measurement, I agree with the authors’ argument that the increased ORAC value is attributable to the presence of selenium in the tumor tissues. It is likely that other tissues or organs from mice treated with Selol would also show increased ORAC values.

Response 2: We appreciate comments on the ORAC measurements and the need for additional quantitative data to support our findings. We agree that the increased ORAC values observed in tumor tissues are likely due to the presence of selenium, and it is reasonable to expect similar increases in other tissues or organs of Selol-treated mice, especially that it was proved in our recent studies on healthy mice. We add it to the Discussion: “The ORAC value, indicating the antioxidant capacity of tumor tissues, was observed to be higher after Selol treatment. This elevated ORAC value is likely associated with the presence of selenium. It is probable that other tissues or organs in tumor-bearing mice treated with Selol would similarly exhibit enhanced ORAC values. Consistent with this, our recent studies in healthy mice have demonstrated significant increases in ORAC levels in both blood and organs following Selol supplementation [29].”

Reviewer 2 Report

Comments and Suggestions for Authors

The authors have adequately addressed all of my concerns. The manuscript has been improved by the revision, I have no further comments.

Author Response

We would like to sincerely thank the Reviewer for their thorough evaluation of our manuscript. Your insightful comments and constructive suggestions have significantly contributed to the improvement of our work. We appreciate the time and effort you have invested in reviewing our article, and we have carefully considered all your feedback in our revisions. Your expertise has greatly enhanced the quality and clarity of our research, and we are grateful for your valuable input.

Kind regards,

Małgorzata Sochacka

Round 3

Reviewer 1 Report

Comments and Suggestions for Authors

The manuscript has shown significant improvement from its initial version. I would support its publication after addressing the minor issue listed below.

Tables 1 to 3 and Figure 1 indicate N=5 in each group. However, Figure 2 shows 6 mice (4+2 and 5+1) in the mentioned groups. On the other hand, Section 2.3, which outlines the experimental protocol, does not clearly specify the number of mice in each group. Could the authors clarify this inconsistency? Additionally, could the author enhance the quality of Figure 2 to improve its presentation?

Author Response

Dear Reviewer,

Thank you for your positive feedback and for pointing out the minor issues that need clarification.

Comment 1: Tables 1 to 3 and Figure 1 indicate N=5 in each group. However, Figure 2 shows 6 mice (4+2 and 5+1) in the mentioned groups. On the other hand, Section 2.3, which outlines the experimental protocol, does not clearly specify the number of mice in each group. Could the authors clarify this inconsistency? 

Response 1: We acknowledge that the initial manuscript did not clearly reflect the correct group sizes in Section 2.3, which led to a discrepancy between Tables 1-3, Figure 2, and the text. To address this issue, we have revised Section 2.3 to accurately specify the number of mice in each group, ensuring consistency across the entire manuscript. The description now explicitly mentions the number of individuals in each group, which should resolve the confusion. We have also made improvements to the captions of the tables (Tables 1-3) and figures (Figure 1-2) for better clarity and consistency throughout the manuscript.

Comment 2:  Additionally, could the author enhance the quality of Figure 2 to improve its presentation?

Response 2: Thank you for pointing this out. The appropriate correction was made.

We hope these revisions will meet your expectations. Thank you once again for your valuable suggestions.

Best regards,

Małgorzata Sochacka
